# Warmstart Approach for Accelerating Deep Image Prior Reconstruction in Dynamic Tomography

**Tobias Knopp**[1,2]                                                            T.KNOPP@UKE.DE
[1] *Section for Biomedical Imaging, University Medical Center Hamburg-Eppendorf, Germany*
[2] *Institute for Biomedical Imaging, Hamburg University of Technology, Germany*

**Mirco Grosser**[1,2]                                                          MI.GROSSER@UKE.DE

**Editors:** Under Review for MIDL 2022

## Abstract

Deep image prior (DIP) has been successfully used in the field of tomography to obtain high-quality images from under-sampled and noisy measurements. The key advantage of DIP compared to conventional deep-learning based image reconstruction techniques is that it requires no training data and thus can be used in a flexible manner without incorporating domain specific knowledge. The downside of DIP is that it shifts the training step to reconstruction time where usually fast algorithms are required to reduced the latency between acquisition and display of the reconstructed image. In this work we tackle this problem for dynamic tomography scenarios in which a large number of temporally resolved images are taken over time. By initializing the DIP network using a previous frame of the time series, it is possible to significantly reduce the overall reconstruction time. To cope with abrupt changes in the captured time-series, we propose to use an adaptive restart method having the ability to switch between warm- and coldstart depending on the amount of inter-frame changes.

**Keywords:** deep image prior, image reconstruction, dynamic tomography, warmstart, magnetic particle imaging

## 1. Introduction

Deep learning (DL) is an important method influencing the field of tomographic image reconstruction. In most cases it provides superior image quality, requires less measurements, and allows for higher noise levels compared to classical approaches utilizing hand-crafted regularization to incorporate prior knowledge of the desired solution. Most DL reconstruction algorithms require training data and either learn the entire imaging operator (Zhu et al., 2018) or just the regularizer (Mukherjee et al., 2020). The latter has the advantage of requiring less training data. Still, the requirement of good training data is problematic since it considerably increases the effort compared to classical image reconstruction approaches.

One very interesting alternative is the deep image prior (DIP) introduced in (Ulyanov et al., 2018). It is an unsupervised learning technique that represents the unknown object using a neural network and optimizes the network parameters instead of the object itself. DIP usually achieves similar image quality as supervised approaches but has the huge advantage of requiring no training data. Originally proposed for image processing, DIP has been successfully used for image reconstruction in various imaging modalities such as

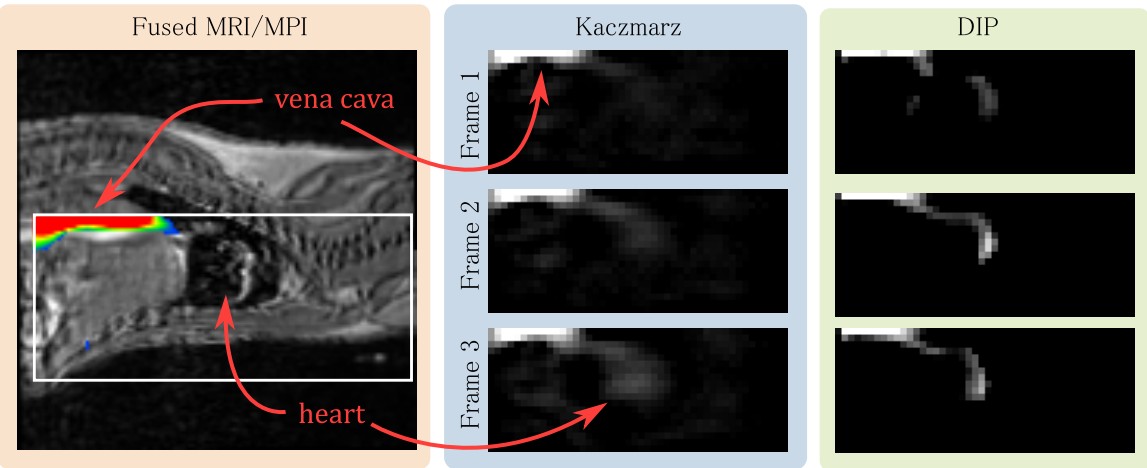

Figure 1: Motivating application scenario for the warmstarted DIP reconstruction. The middle and right column show three frames from a dynamic 3D MPI experiment where a contrast agent bolus is entering a mouse heart via the vena cava. The DIP reconstruction (right) shows much finer resolution compared to the Kaczmarz method (middle). On the left, an MRI/MPI overlay is shown for anatomical reference (see appendix A for experiment details).

positron emission tomography (Gong et al., 2018), computed tomography (Baguer et al., 2020), magnetic resonance imaging (Yoo et al., 2021), and magnetic particle imaging (MPI) (Dittmer et al., 2021).

Despite its large potential for solving tomographic image reconstruction problems, there are some major challenges for DIP reconstruction methods:

1. the architecture challenge
2. the stopping criterion challenge
3. the runtime challenge

The first challenge of the DIP is that its accuracy highly depends on the network architecture being used and systematic approaches for finding an appropriate architecture are scarce. The second challenge of the DIP reconstruction is the need for an appropriate stopping criterion since early stopping is necessary to prevent fitting the model to the measurement noise. The third challenge is that the DIP shifts the network training workload from a pre-measurement phase to the reconstruction phase. This makes it computationally demanding and prevents the use in real-time applications such as interventional imaging (Salamon et al., 2016).

In this work, we further investigate the DIP in the context of MPI. The latter is a functional imaging modality that features a high temporal resolution in the order of 46 frames/s and thus is in strong need of a fast DIP reconstruction. As a motivating example, Figure 3 compares a standard MPI reconstruction (based on the Kaczmarz algorithm (Knopp et al., 2010)) and the corresponding DIP reconstruction. The DIP outperforms the classical reconstruction in terms of spatial resolution and structural clarity. As a matter of fact, many applications (Haegele et al., 2016; Salamon et al., 2016; Herz et al., 2018; Kaul et al., 2018)

of MPI require real-time visualization of the acquired images, which is hard to achieve using current DIP methods.

Based on the previous discussion, this work aims to address the runtime challenge. One approach to reduce the runtime cost is to use an existing DL reconstruction algorithm as the basis for the DIP reconstruction (Barbano et al., 2021). This approach significantly accelerates DIP reconstruction but it requires training data. In this work, we instead target dynamic imaging applications, such as interventional or cardiac imaging, in which a series of frames is imaged. The common way to apply DIP reconstruction to dynamic tomography would be to initialize the network parameters before the reconstruction of each frame, which avoids frame-coupling and preserves temporal resolution. In this work, we instead propose to reuse the network parameters from a previous reconstruction, which can significantly speed-up the convergence of the DIP reconstruction. Our approach is related to (Lei et al., 2020) where network parameters have been reused to achieve consistency in the time series.

## 2. Methods

In dynamic imaging experiments, one measures raw data vectors $\boldsymbol{u}_l \in \mathbb{K}^M$ where $l = 1, \ldots, L$ is the frame index and $\mathbb{K}$ are the real or complex numbers. The measured data is linked to the tomographic object being imaged $\boldsymbol{c}_l \in \mathbb{K}^N$ by the imaging equation

$$\boldsymbol{S}\boldsymbol{c}_l = \boldsymbol{u}_l + \boldsymbol{\varepsilon}_l, \tag{1}$$

where $\boldsymbol{S} \in \mathbb{K}^{M \times N}$ is the imaging operator describing the physics of the imaging process and $\boldsymbol{\varepsilon}_l$ is the noise of the $l$-th measurement. Since the inverse problem (1) is usually ill-posed, the least squares solution

$$\|\boldsymbol{S}\boldsymbol{c}_l - \boldsymbol{u}_l\|_2 \overset{\boldsymbol{c}_l}{\to} \min$$

is generally unstable, due to noise amplification. To cope with this issue, one needs to apply regularization techniques that can be formalized as

$$D(\boldsymbol{S}\boldsymbol{c}_l, \boldsymbol{u}_l) + R(\boldsymbol{c}_l) \overset{\boldsymbol{c}_l}{\to} \min, \tag{2}$$

where $D : \mathbb{K}^M \times \mathbb{K}^M \to \mathbb{R}_+$ is the data discrepancy term and $R : \mathbb{K}^N \to \mathbb{R}_+$ is the regularization term. The former is usually chosen to be the $\ell^1$ or the $\ell^2$ distance while for the latter several terms like $\|\cdot\|_1$, $\|\cdot\|_2$, $\mathrm{TV}(\cdot)$ are used to incorporate prior knowledge about the solution $\boldsymbol{c}_l$. While (2) delivers high-quality result, given a suitable regularizer, these methods often require sophisticated hyper-parameter tuning.

In recent years, various approaches have been proposed to incorporate prior knowledge using deep learning. Here, we consider the unsupervised DIP approach that requires no training data and instead exploits the architecture of a deep neural network to apply regularization. DIP reconstruction can be formulated as

$$D(\boldsymbol{S}\varphi_{\boldsymbol{\theta}_l}(\boldsymbol{n}), \boldsymbol{u}_l) \overset{\boldsymbol{\theta}_l}{\to} \min, \tag{3}$$

where $\varphi_{\boldsymbol{\theta}}$ is a neural network with parameters $\boldsymbol{\theta}$ and $\boldsymbol{n} \in \mathbb{R}^N$ is a noise vector. Optionally an additional explicit regularization term $R(\varphi_{\boldsymbol{\theta}_l}(\boldsymbol{n}))$ can be added (Baguer et al., 2020) but the main tool for regularization with the DIP remains the network $\varphi_{\boldsymbol{\theta}_l}(\boldsymbol{n})$.

In practice, (3) is solved iteratively using an optimizer such as gradient descent or Adam (Kingma and Ba, 2014). In each iteration a function trainingstep is applied which updates the network parameters $\boldsymbol{\theta}_l$ such that the loss decreases. The iteration stops once a stopping_criterion is fulfilled. Algorithm 1 summarizes the tomographic DIP reconstruction for a series of $L$ frames.

---

**Algorithm 1:** Coldstarted DIP Reconstruction

---
**Input:** $D$, $\varphi$, $\boldsymbol{n}$, $\boldsymbol{S}$, $\boldsymbol{u}_l$, $l = 1, \ldots, L$
**Output:** $\boldsymbol{c}_l$, $l = 1, \ldots, L$
1 **for** $l \leftarrow 1$ **to** $L$ **do**
2     $\boldsymbol{\theta}_l \leftarrow \mathsf{randn}(N)$
3     **while** stopping_criterion $=$ false **do**
4         $\boldsymbol{\theta}_l \leftarrow \mathsf{trainingstep}(\boldsymbol{\theta}_l, D, \varphi, \boldsymbol{n}, \boldsymbol{S}, \boldsymbol{u}_l)$
5     **end**
6     $\boldsymbol{c}_l \leftarrow \varphi_{\boldsymbol{\theta}_l}(\boldsymbol{n})$
7 **end**

---

One important aspect of algorithm 1 is that the network parameters are re-initialized after each reconstruction allowing the network to start fresh and not get biased by a previous solution. A drawback of this approach is that a large number of iterations is needed to reach a high quality image. In this work, we propose to instead reuse the network parameters while going from one frame to the next. This is called a warmstart in the field of optimization and leads to algorithm 2.

Note that algorithm 2 uses the warmstart in an adaptive manner, where for each frame a restart_criterion determines whether the previous parameters or a random initialization should be used. This procedure aims to avoid the situation that the optimizer gets stuck in a local minimum when the image changes too severely between frames. One simple restart_criterion capturing inter-frame changes is to restart only when the condition $\frac{2\|\boldsymbol{u}_l - \boldsymbol{u}_{l-1}\|_2}{\|\boldsymbol{u}_l\|_2 + \|\boldsymbol{u}_{l-1}\|_2} > \tau$ holds for a predefined threshold $\tau$. This method has the advantage that the raw data vectors are directly available. Another possibility is to use a cheap reconstruction technique, such as the Kaczmarz algorithm in MPI, to obtain a low-quality reconstruction. With this at hand, the restart_criterion can be defined as $\mathrm{PSNR}(\boldsymbol{c}_l^{\mathrm{cheap}}, \boldsymbol{c}_{l-1}^{\mathrm{cheap}}) < \tau$ where $\boldsymbol{c}_l^{\mathrm{cheap}}$ are the results obtained using the cheap reconstruction algorithm. In this work we compare two different warmstarting approaches. The first is a naive warmstart method where the restart_criterion always returns true. The second one uses an adaptive approach based on the image-based restart_criterion with a threshold of $\tau = 15$.

## 3. Experiments

### 3.1. Study Design

We evaluate the warmstarted DIP reconstruction using two dynamic MPI experiments. The first experiment aims to validate the warmstarted DIP reconstruction and test its limits in a controlled setting. For this purpose, we simulated an experiment featuring a cone (1 mm radius tip, 10 degrees apex angle, and 22 mm height) that is moved through

---

**Algorithm 2:** Warmstarted DIP Reconstruction

---

**Input:** $D$, $\varphi$, $\boldsymbol{n}$, $\boldsymbol{S}$, $\boldsymbol{u}_l$, $l = 1, \ldots, L$

**Output:** $\boldsymbol{c}_l$, $l = 1, \ldots, L$

1 **for** $l \leftarrow 1$ **to** $L$ **do**
2    **if** restart_criterion **then**
3       $\boldsymbol{\theta}_l \leftarrow$ randn$(N)$
4    **else**
5       $\boldsymbol{\theta}_l \leftarrow \boldsymbol{\theta}_{l-1}$
6    **end**
7    **while** stopping_criterion $=$ false **do**
8       $\boldsymbol{\theta}_l \leftarrow$ trainingstep$(\boldsymbol{\theta}_l, D, \varphi, \boldsymbol{n}, \boldsymbol{S}, \boldsymbol{u}_l)$
9    **end**
10    $\boldsymbol{c}_l \leftarrow \varphi_{\boldsymbol{\theta}_l}(\boldsymbol{n})$
11 **end**

---

the field-of-view according to the sequence shown in the first column of Figure 2. After two translations in frames 1–3, we abruptly change the cone in frame 4 into a sphere to evaluate if the warmstarted DIP reconstruction can also handle strong changes in the data stream. Finally in frame 5 the phantom is reverted back to a cone. To put this into perspective, we note that the translations describe realistic dynamics which could also arise in practical experiments. The deformations in frames 3-5 describe a more extreme test case going beyond what is typically observed in MPI experiments. All simulations are performed by forward projection using a measured system matrix. White noise is added to the simulated voltage vector with a standard deviation of 0.2 % of its maximum value. The design of the cone phantom closely follows the cone phantom dataset contained in the OpenMPIData database (Knopp et al., 2020). The latter is a commonly used database for the validation and benchmark of MPI reconstruction methods. Notably it was also used for the validation of the DIP approach in (Dittmer et al., 2021).

The second experiment is an *in-vivo* dataset of a mouse, where MPI was used to image the distribution of the MPI contrast agent after the injection of a bolus (Graeser et al., 2017). This dataset thus features realistic structures and dynamics for a typical application of MPI. For an illustration of the imaging setting, we refer to Fig. 3, which was generated using the described *in-vivo* dataset.

For further details on both experimental setups, we refer the reader to Appendix A, which collects the experimental parameters used in the respective experiments.

### 3.2. Reconstruction & Implementation Details

The architecture of the neural network $\varphi$ is chosen to be an autoencoder without skip connections having approximately 3 million parameters. It has three down-sampling steps by a factor of two with 64, 128 and 256 channels respectively. For more details we refer to (Dittmer et al., 2021). The input data $\boldsymbol{n}$ is initialized randomly with values sampled uniformly from the interval $[0, 0.7]$. In all reconstructions, the same random vector is used. Adam with a learning rate of $5 \times 10^{-4}$ and a standard momentum setting of $\beta = (0.9, 0.999)$

Figure 2: Comparison of coldstarted (green box) and warmstarted (orange box) DIP reconstruction for the phantoms shown in the blue box. Each row corresponds to one frame. Shown are reconstruction results after 1,51,101, 201, and 301 iterations.

is used as the optimizer. The data discrepancy term $D$ is chosen to be the $\ell^1$ distance, which has been shown in (Kluth and Jin, 2020) to be advantageous compared to an ordinary $\ell^2$ distance in MPI. We note that all settings are chosen similarly to the ones proposed in (Dittmer et al., 2021), such that we rely on a well-tested setting.

In all experiments we use a static number of 400 iterations as the stopping_criterion, which makes it easier to compare cold- and warmstarted DIP reconstructions. In practice one would base the stopping criterion on the quality of the image.

The algorithms and the evaluation study are implemented in the programming language Julia (Bezanson et al., 2017). MPI related parts use the packages MPIFiles.jl (Knopp et al., 2019a) and MPIReco.jl (Knopp et al., 2019b). The DIP reconstruction is implemented using the package Flux.jl (Innes, 2018; Innes et al., 2018) and runs on a GPU (Nvidia Quadro M4000).

## 4. Results

Reconstruction results for the simulation study are shown in Figure 2. One can see that the coldstarted DIP reconstruction requires about 100 iterations to reach a level where the result matches the phantom. The reconstruction result is not perfect, since the noise in the measurements is high and represents a realistic case for an actual MPI experiment. It can also be seen that the convergence behavior is qualitatively similar for all frames, which is due to the fact that the phantom is similar in all frames and the network is always restarted with fresh parameters.

Table 1: Number of necessary iterations to reach an PSNR of 23 for the cold- and the warmstart method.

| Method | Frame 1 | Frame 2 | Frame 3 | Frame 4 | Frame 5 |
|---|---|---|---|---|---|
| coldstart | 89 | 91 | 88 | 200 | 88 |
| warmstart | 91 | 10 | 11 | 162 | 64 |

Turning to the naively warmstarted DIP reconstruction one observes that the result after the first iteration looks basically the same as the final result of the previous frame. In the following iterations, the phantom is quickly morphed to the new position such that a satisfying results is already obtained after less than 51 iterations. An exception to this behavior arises in frames four and five. Here one observes that the DIP requires more iterations to deal with the strong object changes.

To quantify the observations made before, Table 1 shows the number of iterations required by the two different starting strategies to reach an PSNR of 23. According to this metric, the coldstarted method requires between 88 and 200 iterations to achieve a satisfying result. On the other hand, the warmstarted method achieves an PSNR larger than 23 in just 10 respectively 11 iterations for frames two and three. Even for the frames four and five with the abrupt phantom change, the warmstarted DIP achieves slightly better results than the coldstarted DIP. More details concerning the convergence of both methods can be found in appendix B.

Finally, we turn to the adaptively warmstarted DIP reconstruction, which chooses whether to warmstart based on the estimated strength of inter-frame changes. In Figure 2, the method chosen by the adaptive method is highlighted by a red box for each frame. As one can see, the first three frames are all reconstructed using warmstart, whereas the remaining frames featuring strong object deformations are reconstructed without warmstart, thus minimizing computation time.

To show the practical relevance of the proposed methods we applied it to the experimental *in-vivo* data outline in Figure 1 and appendix A. The results are shown in Figure 3. As can be seen, the coldstarted DIP requires about 200 - 400 iteration to reach a satisfying result. In contrast, the warmstarted DIP requires only 50 - 100 iterations marking an acceleration by roughly a factor of four.

## 5. Discussion

Our results show that a warmstart can significantly accelerate the convergence speed of the DIP in dynamic image reconstruction applications. In cases where the object position changes only slightly, the network parameters can be adapted quickly by just few optimization steps. For larger inter-frame distances the optimization needs longer since the network parameters need to be change more in order to adapt to the new object position and shape.

We also highlighted challenges of the warmstarted DIP reconstruction, which is the case when the object changes abruptly. But even in this case, the warmstarted DIP did not result in worse convergence behavior for the phantoms used in our experiment. Nevertheless we

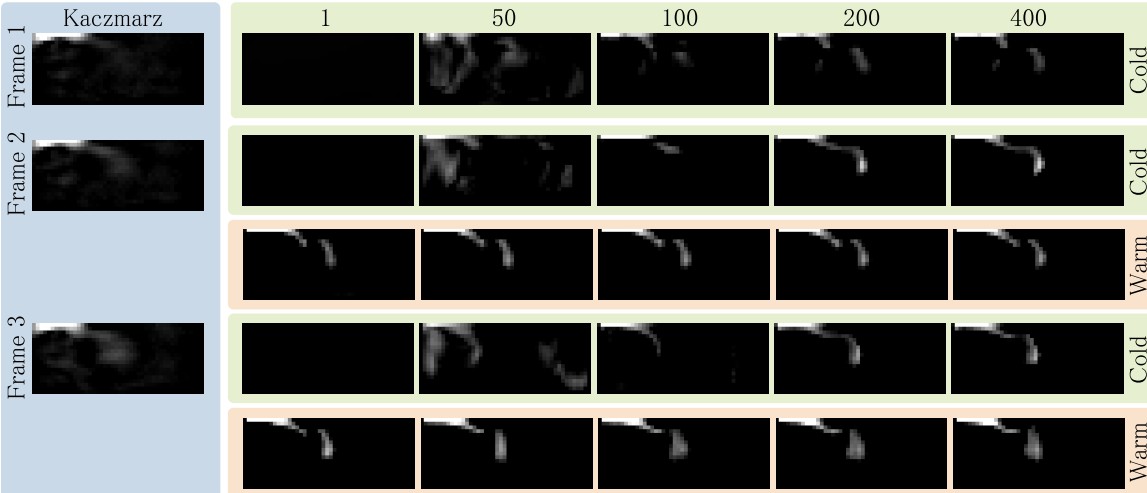

Figure 3: Comparison of coldstarted (green boxes) and warmstarted (orange boxes) DIP reconstruction of the *in-vivo* MPI experiment showing the bolus inflow into a mouse heart. For reference, the blue box shows a conventional reconstruction using the Kaczmarz algorithm.

propose to use an adaptively warmstarted DIP reconstruction since the bias introduced by warmstarting should be used with care only in situations where the object changed slightly. In Figure 2 we used a simple criterion based on the difference between subsequent Kaczmarz reconstructions and chose a conservative threshold of $\tau = 15$ to decide on the restart. This values was tailored towards the concrete experiment and further investigations are necessary for automatic selection of this hyperparameter.

An advantage of the proposed method is its high simplicity making it applicable in a wide range of applications. As an alternative to the sequential approach used here, (Yoo et al., 2021) propose a time-dependent DIP which performs a joint reconstruction of the full image series. While yielding an efficient method with high-quality results, the joint-processing of all frames makes it hard to realize real-time visualization, which is needed for some MPI applications.

In conclusion, the proposed warmstart method is a simple, yet effective tool for the acceleration of DIP reconstructions. In particular, we demonstrated that the warmstart method is well-suited for the reconstruction of dynamic MPI image series. This can be viewed as a first step towards a DIP method providing real-time visualization in MPI.

## Acknowledgments

We thank Sören Dittmer and Johannes Leuschner for many helpful discussions on DIP image reconstruction.

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

## Appendix A. Experimental Parameters

Generation of the simulated MPI data was performed using a system matrix taken from the OpenMPIData (Knopp et al., 2020) project (calibration dataset 3). It was measured

with a preclinical MPI scanner (Bruker, Ettlingen) applying a 3D measurement sequence capturing a volume of $24 \times 24 \times 12$ mm$^3$ using a gradient strength of 1 Tm$^{-1}$ in the $xy$-directions and 2 Tm$^{-1}$ in $z$-direction. The system matrix was sampled in a slightly larger volume ($32 \times 32 \times 16$ mm$^3$) on a grid of size $16 \times 16 \times 16$. The repetition time for capturing a volume was $21.54\,\mu s$ such that 46 volumes were measured per second. From the measured system matrix we use only those rows having a signal-to-noise ratio larger 2.0, which is a common strategy for MPI reconstruction algorithms. Based on this system matrix, measured data was simulated by applying the system matrix to the image vector at each time frame. To simulate measurement noise, white noise with a standard deviation of 0.2 % of the maximum voltage signal was added to the simulated voltage vector. While the reconstructed data is fully 3D, the images in Figure 2 and Figure 5 show only a central $xy$ slice through the phantoms.

The in-vivo experimental data is taken from (Graeser et al., 2017). It was acquired with the same 3D sequence used in the simulation study. The only difference is that the system matrix is sampled in a volume of $32 \times 25 \times 13$ mm$^3$ on a finer grid of size $46 \times 36 \times 19$ (isotropic 0.7 mm voxel size). In contrast to the simulation study, the measured *in-vivo* data is shown as a maximum intensity projection along the $y$-direction (sagittal view). The data shows the inflow of the MPI contrast agent injected intravenously into the heart of a living mouse. From the entire imaging experiment we extracted five subsequent frames and performed Kaczmarz reconstruction, coldstarted DIP reconstruction and warmstarted DIP reconstruction (non-adaptive). From the five reconstructed frames only every second is illustrated in Figure 3.

## Appendix B. Details for Simulation Study

To study in more detail the reconstruction result as a function of the iterations, Figure 4 shows the residual $D(\boldsymbol{S}\varphi_{\boldsymbol{\theta}}(\boldsymbol{n}), \boldsymbol{u})$ and the peak signal-to-noise ratio (PSNR) of the frames $3 - 5$. One can see that the coldstarted DIP reconstruction requires about 65 iterations during which both the residual and the PSNR remain static since the network outputs a vector close to a zero vector. Then in iterations $65 - 110$ the image converges rapidly to the desired solution and remains static afterwards. We don't see an overfitting in the first 400 frames, which might be due to the architecture having a limited complexity. For the warmstarted DIP reconstruction one can see that the PSNR quickly increases within the first 10 iterations.

When switching from the cone to the sphere phantom in frame 4 one can see that the warmstarted DIP requires much more iterations for strong object changes. The final plateau is reached after about 200 iterations similar to the coldstarted case. Still for both frames four and five the warmstarted DIP performs better than the coldstarted DIP for the phantom being used in the simulation study.

To study in more detail the impact of the inter-frame similarity on the convergence of the warmstarted DIP reconstruction we repeat a simulation where the position of the cone at frame three of the previous experiment is taken as a reference position and then three different networks are pre-trained with the cone shifted to $2\,$mm, 4 mm and 6 mm. The results are shown in Figure 5. One can see that for all distances the phantom is shifted to the new position after about 20 iterations. However, the shape of the phantom is deformed for

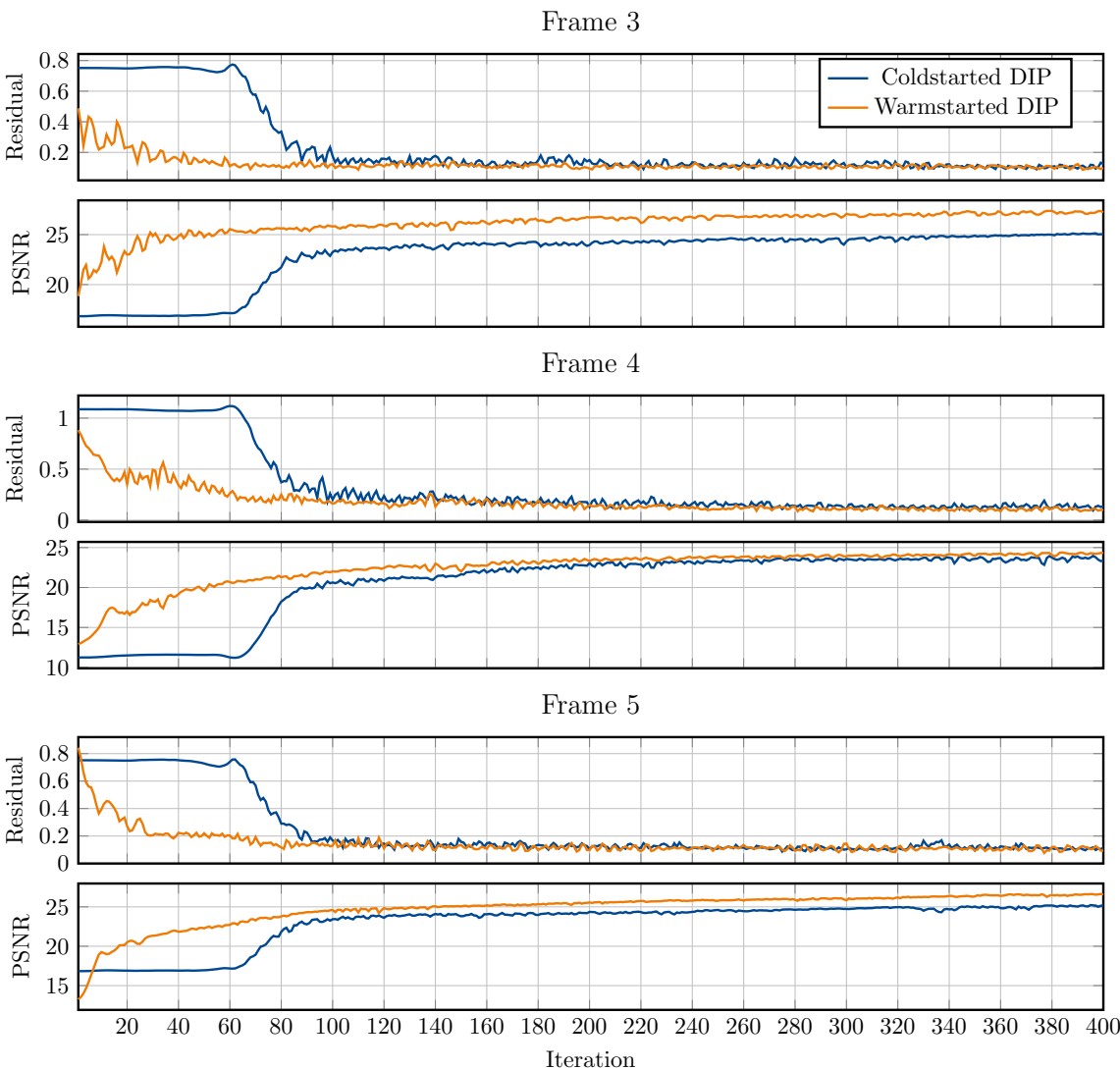

Figure 4: Residual (arbitrary unit) of the data discrepancy term and PSNR of the reconstruction result compared to the true solution for frames $3-5$, both parameter initialization methods, and for all 400 iterations.

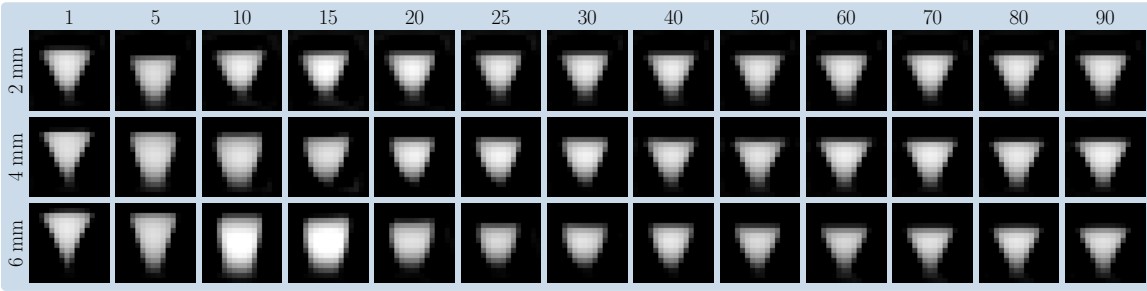

Figure 5: Warmstarted DIP reconstructions for the cone phantom and different phantom distances from frame to frame. Shown is always the same frame, which means that the three rows only differ in the inter-frame distance to the previous frames, which is chosen to be 2 mm, 4 mm, and 6 mm. Reconstruction results of the warmstarted DIP reconstruction are shown after 1, 5, 10, 15, 20, 25, 30, 40, 50, 60, 70, 80, and 90 iterations.

the 4 mm and the 6 mm distance. Thus the 4 mm distance requires about 30–40 iterations while the 6 mm distance requires about 50 iterations to match the image quality obtained after 20 iterations for the 2 mm distance warmstart.

