# OpenReview forum: "Warmstart Approach for Accelerating Deep Image Prior Reconstruction in Dynamic Tomography"
_MIDL.io/2022/Conference — MIDL 2022_

### Official Review · Reviewer_xiTA · 2022-01-14

**Confidence:** 4
**Preliminary Rating:** 4
**Recommendation:** Poster

**Summary:**

The work proposes a warmstart approach which helps in initialization of the weights and hence improves the reconstruction time for the dynamic tomography scenarios. It also shows a comparison with the coldstart method and shows that the times are improved considerably using the deep image prior in the field of dynamic tomography..

**Strengths:**

The strength of the paper lies in the application of warmstart approach for improving the reconstruction time using deep image priors for dynamic tomography. A comparison with the coldstart approach has also been shown which clearly shows the benefits of using such an approach.

**Weaknesses:**

Since it is a methodological paper I will assume that there will be new insights into the method developed. Using a good initialization is very common for getting faster solutions, how can this impact the other scenarios where other techniques for initialization are used.

**Deanonymize Review:**

no

**Detailed Comments:**

1. There is a problem with finding the best architecture and the stopping criteria always for the DL models. How is it different here in this case. Please clarify.

2. Page 3 mentions the use of Equation 3 before defining it, I believe it should be equation 2.

3. As mentioned that the architecture will influence the time taken for reconstruction. How the different strategies work for different type of architectures. Any insights in this will be helpful.

4. The strategy of using a warmstart is the most obvious technique which can be used, can you please show how it is better from some pre-defined initialization for the models.

5. Figure 3 caption repeats the phrase - three rows, 2 times. Please proofread it and check these mistakes.

**Final Rating After The Rebuttal:**

4: Weak Accept

**Justification Of The Final Rating:**

I am satisfied with the revisions from the authors but I feel there should be more experiments with new datasets and comparison with the other methods which will strengthen the work.

In the current revised manuscript, I cannot raise my rating of the work.

**Paper Type:**

methodological development

**Questions To Address In The Rebuttal:**

1. The phantom results are shown for very simple phantoms. How will it translate to a complex phantom.

2. Figure 3 shows the results for warmstarted reconstruction but not for the coldstart reconstruction. Any particular reason behind it. Please explain this or include the results for the coldstart reconstruction also.

3. Please provide more details of the architecture used and why this architecture was chosen. Are there batch normalization in the architecture ? Also there are other strategies for initializing the weights. It will be more beneficial if the method can be shown to perform better than other initialization techniques.

4. Please also explain the contributions. The warmstart seems to be a natural strategy, so if there are more insights or analysis into that, it will be easy to understand the benefits of the proposed warmstart approach.

**Special Issue:**

no

---

### Official Review · Reviewer_jyxV · 2022-01-17

**Confidence:** 5
**Preliminary Rating:** 2
**Recommendation:** Poster

**Summary:**

The paper proposes a warmstart approach for DIP reconstruction in the case where one tries to reconstruct highly correlated samples between the time frames. I find the method and the experiments overly simplistic, and does not add much value to the existing literature in its current form. In order to facilitate the impact, the authors should perform much more experiments, and also compare with the existing works.

**Strengths:**

The method is sound, and simple to implement, probably within 1-2 lines of code. Warmstarted version of DIP shows clear advantage over Coldstarted version, which is expected. The paper is rather easy to follow and understand.

**Weaknesses:**

1. The experimental data is so simple that it is hard to verify if the method will actually be valid on real-world datasets. The authors should try to implement the idea using more data that contain more complicated structure, e.g. in vivo data.

2. The only methodological comparison the authors provide is the comparison with coldstarted DIP. Adding to this comparison, the authors should also consider comparing with the existing approach for using DIP for dynamical imaging [1]. Yoo et al. show that it is indeed unnecessary to train DIP on individual timeframe. What advantage does the proposed method have over [1]?

3. Figure 2 is not really convincing. There is only a single frame where the warmstarted version shows better performance than the coldstarted version once the optimality is reached. However, for Frame 5, the convergence is actually faster when using the coldstarted DIP. The results do not seem robust, and this may be correlated with the concern that I have with the overly simplistic data.

Reference
[1] Yoo et al. "Time-dependent deep image prior for dynamic MRI", IEEE TMI, 2021.

**Deanonymize Review:**

no

**Final Rating After The Rebuttal:**

4: Weak Accept

**Justification Of The Final Rating:**

The paper proposes a clear and concise method for strengthening the method of DIP reconstruction for MPI. Results show that much faster convergence can be achieved, both in the simulation, and the in-vivo dataset. My decision is to accept.

**Paper Type:**

validation/application paper

**Questions To Address In The Rebuttal:**

I would like to advise the authors the following:

1. Add an experiment with another dataset that contains complex anatomy.

2. Show that their method is superior, or at least in par with [1].

3. Work on improving the clarity that the warmstarted version is actually better than the coldstarted counterpart.

Reference
[1] Yoo et al. "Time-dependent deep image prior for dynamic MRI", IEEE TMI, 2021.

**Special Issue:**

no

---

### Official Review · Reviewer_Nahv · 2022-01-23

**Confidence:** 4
**Preliminary Rating:** 1

**Summary:**

This paper proposes a warmstart strategy for DIP network initialization for MPI reconstruction. In particular, for a series of MPI scans, this paper proposes to initialize each subsequent image’s reconstruction network weights with the learned weights of the previous scan. The goal is for this strategy to reduce the number of DIP optimization steps required to optimize each image. This strategy is tested on a sequence of 5 phantom MPI scans comprised of a triangle which is shifted twice, transformed to a circle, and transformed back to a triangle. The proposed initialization approach improves reconstruction time for 3 of the 4 frames for which it is used (two of which are translations from the previous frame), and increases reconstruction time for the other frame.

**Strengths:**

- **The paper is well-written and easy to understand.** The exposition on DIP is very clear and would be easy for a newcomer to the field to read and understand.
- **The idea behind this paper suggests an interesting question**: when should we start off with more specific priors on the image than those provided by DIP, and how detailed should they be? The discussion section contains interesting hypotheses toward the first question (I'd be really interested in a paper which developed a good similarity metric in the raw measurement space). For the second, I'm curious, for example, if applying the warm-starting approach developed here only to a subset of layers could improve the quality of the reconstructions. With this strategy, for more complex images, the networks could in theory share basic information about low-frequency structures which are unlikely to change from image to image, while allowing finer-grained structure to be reconstructed from scratch.

**Weaknesses:**

- **The experiments are performed on only one series of 5 scans with very simple phantoms, which cannot shed light on how well this method performs for more realistic images.** For more realistic anatomical images, the relevant differences are defined by finer scale image features; these will likely require a higher SSIM threshold to declare an image adequate. In that setting, a worry is that the warm-started networks get “stuck” close to the initialization, which provides a reasonably low value for the data discrepancy term to begin with, but are unable to move out of that optimum to the one corresponding to the actual acquired image. Overall, given that the method is quite straightforward and does not introduce a new conceptual idea, more sophisticated experiments on real data are needed to demonstrate that this method makes a big difference.

- **The proposed method increases the number of iterations required for one of the two frames where the shape of the reconstructed phantom changes qualitatively from the previous frame**. It is hard to argue that this method is working well since it can’t handle this case. The directions described in the “Discussion” for deciding when to apply a warm-starting method are interesting and seem to be a good starting point for future work to improve the reliability and usability of this method.

**Deanonymize Review:**

no

**Detailed Comments:**

- pg. 3 “The former is usually chosen to be the l1 or the l2 distance while for the later several terms…” —> “The former is usually chosen to be the l1 or the l2 distance, while for the latter, several terms…”
- pg. 3 “When choosing an appropriate regularizer, (3) delivers high quality…”: I believe the should reference Eq. (2).


**Final Rating After The Rebuttal:**

2: Weak Reject

**Justification Of The Final Rating:**

The authors made substantive changes to the paper to address most of my concerns beyond one major one: the method is tested only on two sequences of images, one of which is simulated phantom data.

This is problematic because (1) there is no characterization of how robust/generalizable the method is and (2) the new warmstart formulation includes a hyperparameter, and we need to see how this performs across held-out images. As mentioned in my comments below, I encourage the authors to work further in the future to better understand how this method performs on a larger number of more varied images.

**Paper Type:**

methodological development

**Questions To Address In The Rebuttal:**

- **Are there any MPI use cases requiring realtime visualization and adjustment, such that the 46 frames/s acquisition rate also applies to the reconstruction?**  While I understand why the frames are _acquired_ in rapid succession, I do not understand why they must be reconstructed in rapid succession — could these not be reconstructed after the frames are acquired? If there is a realtime use case, it would be helpful to explicitly specify this use case in the introduction. In any case, it would be useful to report results in terms of reconstruction time (ms) instead of iteration numbers, to put the speed-ups/slowdowns required here in the context of the goal reconstruction rate (or, if the benefit is to the practitioner, to understand how much time a user would be able to save).
- **Are there any practical use cases of MPI where the subject being imaged does not have any fine-grained features?** This would change my view on how relevant the experiments are, although in this case I would still need to see performance on a larger collection of images to understand how reliable the method is.

**Special Issue:**

no

---

### Meta-Review · Area_Chair_R2Uw · 2022-02-16

**Recommendation:** Accept (Poster)
**Confidence:** 4

**Metareview:**

The paper has received three detailed reviews and enjoyed a lively discussion during review/response. The central criticism of all reviewers prior to revision was the simplicity of the data used to demonstrate the method as well as the limited choice of baselines. After response, especially due to the inclusion of an experiment on in-vivo data, all reviewers have raised their review scores, skewing marginally towards acceptance.

---

### Decision · Program_Chairs · 2022-02-28

Accept